# Optimizing Crop Yield Estimation through Geospatial Technology: A Comparative Analysis of a Semi-Physical Model, Crop Simulation, and Machine Learning Algorithms

**Murali Krishna Gumma** [1,*] **, Ramavenkata Mahesh Nukala** [2] **, Pranay Panjala** [1] **, Pavan Kumar Bellam** [1] **, Snigdha Gajjala** [1] **, Sunil Kumar Dubey** [3] **, Vinay Kumar Sehgal** [4] **, Ismail Mohammed** [1] **and Kumara Charyulu Deevi** [1]

1   Geospatial Sciences and Big Data, International Crops Research Institute for the Semi-Arid Tropics (ICRISAT), Hyderabad 502324, India
2   Faculty of Geo-Engineering, Andhra University, Visakhapatnam 530003, India; nukalma@yahoo.com
3   Mahalanobis National Crop Forecast Centre, Delhi 110012, India
4   Indian Agricultural Research Institute, Delhi 110012, India
*   Correspondence: muralikrishna.gumma@icrisat.org

**Abstract:** This study underscores the critical importance of accurate crop yield information for national food security and export considerations, with a specific focus on wheat yield estimation at the Gram Panchayat (GP) level in Bareilly district, Uttar Pradesh, using technologies such as machine learning algorithms (ML), the Decision Support System for Agrotechnology Transfer (DSSAT) crop model and semi-physical models (SPMs). The research integrates Sentinel-2 time-series data and ground data to generate comprehensive crop type maps. These maps offer insights into spatial variations in crop extent, growth stages and the leaf area index (LAI), serving as essential components for precise yield assessment. The classification of crops employed spectral matching techniques (SMTs) on Sentinel-2 time-series data, complemented by field surveys and ground data on crop management. The strategic identification of crop-cutting experiment (CCE) locations, based on a combination of crop type maps, soil data and weather parameters, further enhanced the precision of the study. A systematic comparison of three major crop yield estimation models revealed distinctive gaps in each approach. Machine learning models exhibit effectiveness in homogenous areas with similar cultivars, while the accuracy of a semi-physical model depends upon the resolution of the utilized data. The DSSAT model is effective in predicting yields at specific locations but faces difficulties when trying to extend these predictions to cover a larger study area. This research provides valuable insights for policymakers by providing near-real-time, high-resolution crop yield estimates at the local level, facilitating informed decision making in attaining food security.

**Keywords:** crop yield; DSSAT; ML algorithms

## 1. Introduction

Accurate information regarding the productivity of staple crops at their scale is highly essential for successful national planning as well as for ensuring food security at the country level. The application of satellite-based remote sensing emerges as a practical and cost-effective strategy for thorough crop monitoring, both at regional and national levels [1]. The agricultural sector plays an important role in India's economy, contributing almost 19.9% to the nation's GDP and employing almost half of the country's workforce [2]. Wheat is one of the major staple crops grown during the post-rainy season in India, especially in Eastern India. It is cultivated with broad adaptability from temperate to cold environments in Northern India. The major wheat-producing states in India are Uttar Pradesh, Madhya Pradesh, Punjab, Haryana and Bihar [3].

Accurate wheat crop yield predictions are essential for farmers to better plan their production as well as their participation in the international wheat trade [4]. Advance estimations of crop productions are crucial for policymakers, as they enable them to prepare for crop procurement, distribution, determining price structure and strategizing import/export decisions [5]. For farmers, this helps to determine their optimum area allocation under different crops and ensure they can maximize their production and income [6]. Recent advancements in technology, data collection and computational efficiency have facilitated the design and implementation of big-data analytical approaches, which involve the use of historical crop data, satellite imagery, climate data and other relevant information to build a model that can forecast crop production at a given point in time.

Geospatial technology has good potential for sustainable agricultural development, environmental assessment, assessing crop suitability, monitoring cropland changes, etc. [7–11]. Numerous studies have focused on monitoring land use and land cover (LULC) using a variety of satellite imagery resolutions and techniques, spanning from local to global scales [12,13]. Remote sensing has proven to be a highly effective method for tracking the spatial distribution of agricultural croplands and LULC classes [14,15]. The use of satellite-based spatial–temporal imagery enables the rapid, extensive, cost-effective and continuous monitoring of crop fields [16]. Several studies have mapped LULC and crop types using machine learning (ML) algorithms to classify satellite data, such as Sentinel-2 imagery [17–19] and semi-automated techniques for crop types [20]. While ML techniques have proven valuable in categorizing croplands, they necessitate a substantial quantity of high-quality training data [21–23]. Unsupervised methods like spectral matching techniques (SMT) can perform well with good accuracy with minimal ground data [24].

Remote sensing has widespread applications globally in estimating crop yields. Remote sensing applications were initially focused on classification themes, which involved identifying and mapping objects [18]. The estimations derived from remote sensing commonly rely on vegetation indices (VIs), employing a simple regression against the leaf area index (LAI) or the fraction of Photosynthetically Active Radiation (fAPAR) [25–27]. These estimates have demonstrated a high level of reliability and concordance with actual harvest data. However, it is crucial to emphasize the need for local calibration to ensure the precision of the data.

The crop growth model has been successful in simulating the growth and behavior of real crops since the 1960s [28]. Several crop models have been developed to study the relationship between soil, plants and the atmosphere and to estimate biomass and grain yield [29]. These mathematical equations have undergone pre-training using a diverse range of experimental data gathered from various environments. Subsequently, they are fine-tuned or calibrated to enhance precision in predicting outcomes within specific studies [30]. While these mathematical models exhibit reasonable prediction accuracy, their practical applicability is hindered by the need for extensive calibration, prolonged runtimes and constraints related to data storage [31,32]. Since they are typically built on a regional scale, they often overlook significant spatial and temporal variations in soil characteristics, crop parameters and meteorological data, leading to uncertainties that affect the model's accuracy [33]. The integration of remote sensing data addresses these issues by offering real-time acquisition and spatial continuity, enhancing the applicability of a crop growth model at a regional scale. Integrating the advantages of both remote sensing and a crop growth model helps mitigate uncertainties and improve accuracy in representing the physiological growth process, supporting better agricultural planning. Many studies have attempted to utilize this strategy of combining crop modelling and remote sensing [34,35]. Past research studies on wheat yield estimation have combined remote sensing with various crop models [36], including WOFOST [37,38], CERES-Wheat [39–41], Decision Support System for Agrotechnology Transfer (DSSAT) [42], APSIM, INFOCROP [43], STICS [44], SAFY [45], Wheat-Grow [46], etc.

The DSSAT is known for its standardized input format, which permits consistent data entry. It has undergone extensive validation studies, contributing to its credibility.

The DSSAT is versatile, covering a wide range of crops with a common shell to manage input data and the crop model. The DSSAT-CERES-Wheat model is part of the DSSAT-CSM [47,48]. The model simulates the growth and development of cereal crops, such as wheat, incorporating weather and management variables. Demonstrating its effectiveness, the model serves as a valuable tool for identifying agricultural production management practices that effectively mitigate economic uncertainties. The CERES-Wheat model has successfully simulated wheat growth and yield in response to various conditions, such as soil and climate conditions, in multiple studies [49–51].

The integration of remote sensing and a semi-physical model (SPM) for yield estimation has been the subject of numerous studies [52,53]. An SPM incorporates factors such as Photosynthetically Active Radiation (PAR) and the fraction of PAR absorbed by a crop, providing real-time insights into a crop's growing conditions throughout a season. While this approach has shown satisfactory results for estimating potential crop yields for various crops, it provides a potential yield rather than an actual one. The model operates under the assumption of constant radiation conversion efficiency and harvest index, overlooking factors such as nitrogen deficiency influenced by water stress and temperature stress during reproductive and grain-filling stages. While nitrogen deficiency is addressed through fAPAR, the primary challenges affecting the accuracy of yield estimations are water and temperature stress [54].

This present study aimed to compare the effectiveness of using remote sensing in conjunction with ML, DSSAT and SPM in estimating crop yields. The major goal of this study is to map wheat growth areas in the study area, followed by location-specific yield estimation using ML, DSSAT and SPM, and compare their findings.

## 2. Materials and Methods

### 2.1. Study Area

Bareilly district in Uttar Pradesh, situated on the Ganges River plain between 28°8′ and 28°58′ north latitude and 78°58′ and 79°47′ east longitude, experiences a monsoon-influenced climate with an average annual precipitation of 800–900 mm [55]. The district's temperature variations, ranging from hot summers exceeding 40 °C to cool winters between 4 and 20 °C, contribute to the cultivation of a diverse range of crops. The region's fertile alluvial soils support its thriving agricultural sector, with wheat, rice, sugarcane, pulses, oilseeds and vegetables being the primary crops. With the adoption of modern agricultural practices and irrigation facilities, including canals and tube wells, Bareilly district plays a crucial role in Uttar Pradesh's agricultural landscape and agricultural production.

Ground data were gathered throughout the study area in January 2021, as shown in the figure, encompassing all significant crops. These ground data serve the purpose of crop classification and provide insights into the cropping patterns in the study area. Our proprietary mobile application, "iCrops", is employed for collecting this ground data, which includes information such as geographical location, LULC and more. Additionally, Crop Cutting Experiments (CCEs) are conducted across the study area, as illustrated in Figure 1, to obtain details such as geographical location, i.e., latitude and longitude, biomass and weight, for the identification of the harvest index.

### 2.2. Methodology

Mapping Wheat-Growing Areas in the Study Area by Integrating Sentinel-2 Imagery and Ground Data

Sentinel-2 satellite imagery from the Copernicus Program is used in this study for precise crop classification and monitoring. Sentinel-2 data have a high spatial resolution ranging from 10 to 60 m and the availability of 13 spectral bands. This enables us to capture complex details about various crops, enhancing our ability to distinguish and analyze agricultural fields. Additionally, the satellite's notable 5-day revisit time at the equator proves helpful for closely monitoring the growth cycles of crops.

The study combines Sentinel-2 data with ground survey data to create a detailed and accurate crop type map for the rabi season (Figure 2). The methodology correlates the ground signatures obtained from ground data with class signatures [23].

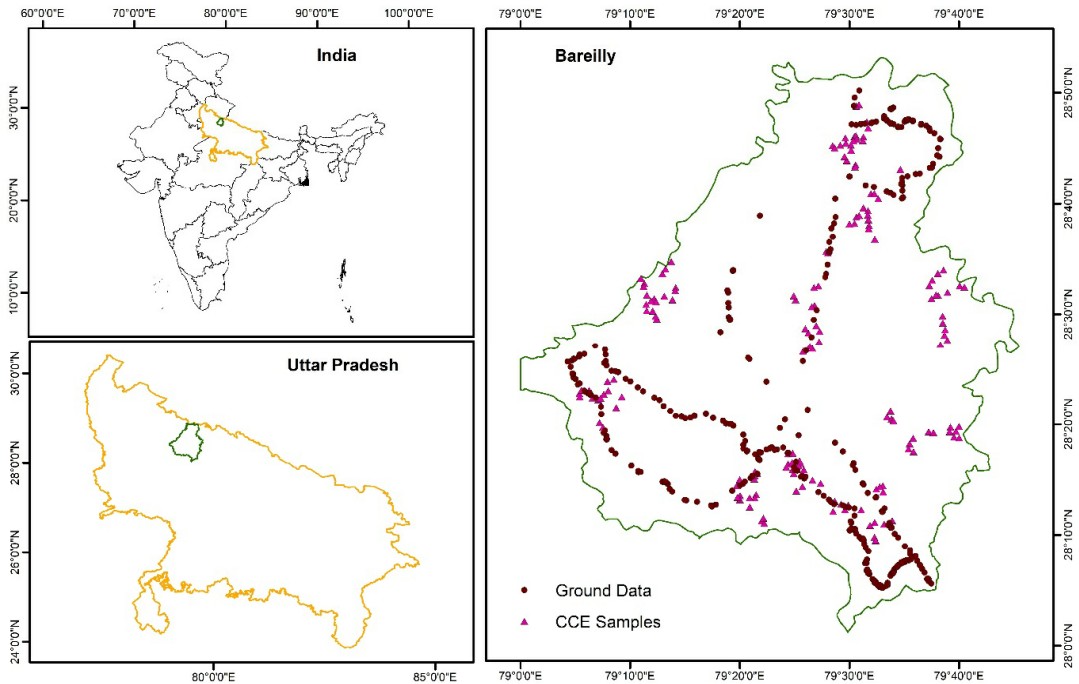

**Figure 1.** Ground points and CCEs across the study area.

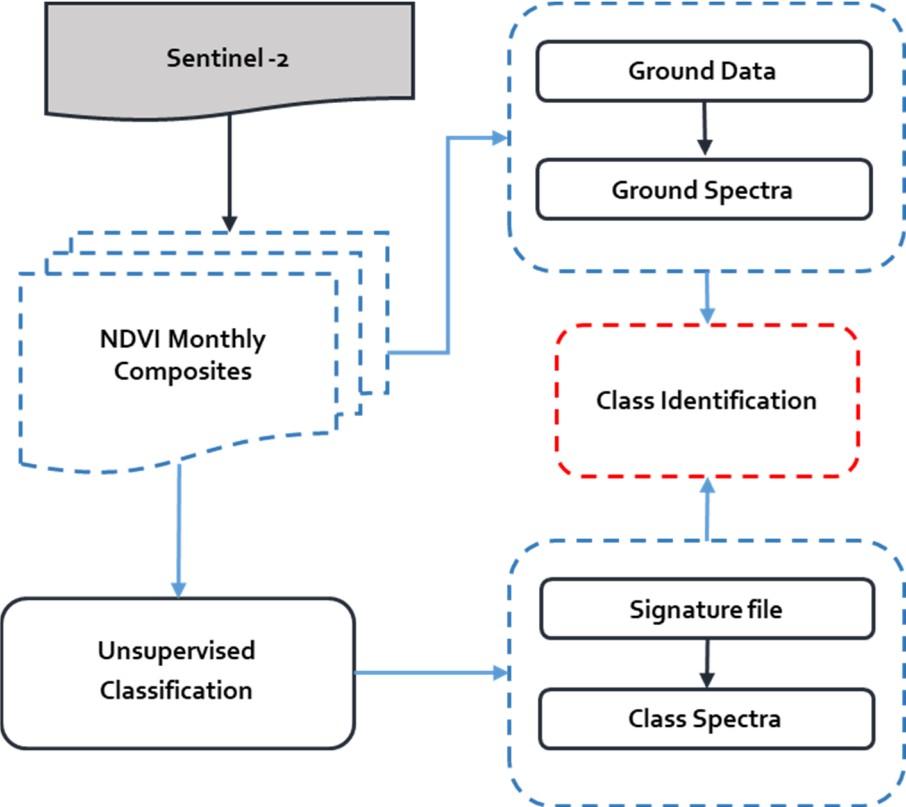

**Figure 2.** Methodology adopted for crop type mapping.

In the collection of ground signatures, we conducted thorough ground surveys to gather precise information about croplands, taking into account essential factors like crop type, the availability of irrigation facilities, soil type, etc. To ensure optimal identification capabilities, we carefully chose homogenous areas, each covering a minimum of $100 \times 100$ m. This selection process enhanced our ability to accurately characterize and understand the composition of these areas. Subsequently, we organized the collected reference samples into similar classes based on critical factors such as cropping intensity, specific crop types and the systems used for cropping.

For the generation of class signatures, we employed an unsupervised classification approach on Sentinel-2 time-series data from the rabi season 2020–2021, i.e., November 2020 to March 2021, utilizing the K-means cluster algorithm. This method enabled us to automatically categorize the data into 100 distinct classes based on their spectral characteristics. Subsequently, we created detailed spectral profiles for each of these identified classes. These profiles provide in-depth information about the unique spectral properties of each class, contributing to a more nuanced and precise understanding of crop categorization.

In the process of matching ground signatures with class signatures, we first grouped the 100 classes into sub-groups based on their spectral profile similarity. This grouping helped us organize and compare the spectral characteristics of different classes efficiently. Later, we established matches between the class signatures and the ground reference samples obtained during the ground surveys, ensuring similarity with factors such as crop type, irrigation facilities, soil type, etc. To enhance the accuracy of the matching process, we set criteria for spectral correlation similarity. If a match meets the established criteria, the class is considered and aligned with the ground truth; otherwise, it retains its assigned name. In cases of mismatches, we masked out those classes for further reclassification. This iterative approach ensures the reliability of the classification results and refines the accuracy of our crop type map.

*2.3. Crop Yield Estimation Using Different Approaches*

(a)    Using Machine Learning Algorithms

Using the Sentinel-2-based Normalized Difference Vegetation Index (NDVI) as a proxy to estimate yield by correlating it with pixel-level CCEs is a common practice in agriculture [53]. The NDVI is a reliable proxy for crop yield as it reflects the health and growth of vegetation. Derived from remote sensing data, it correlates with factors like chlorophyll content and leaf area index, providing insights into crop development. Its non-destructive and cost-effective nature, along with its high spatial and temporal resolution, makes the NDVI a valuable tool for monitoring large agricultural areas. It helps identify stressors and diseases, enabling timely interventions. However, for more accurate predictions, it is essential to integrate NDVI data with other contextual information, such as weather and soil conditions. This process typically involves the NDVI, which is calculated using the near-infrared and red bands of the electromagnetic spectrum.

$$NDVI = (NIR - RED/NIR + RED)$$

CCEs involve physical measurements of crop yield in specific areas of a field. These experimental results are correlated with NDVI values extracted from satellite imagery at the same locations. A random forest machine learning algorithm was employed in this study, utilizing a stack of NDVI layers as input features and a CCE as the training dataset. This approach uses the random forest algorithm to effectively analyze the complex relationships within the stacked NDVI layers, enabling accurate predictions based on the provided CCE training data.

(b)    DSSAT Crop Simulation Model

In this study, the CERES-Wheat model, embedded in the DSSAT v4.7.5 framework, was employed to simulate daily crop growth and development for wheat. Input data, encompassing weather, soil, cultivar/genotype and crop management parameters, were

collected from diverse sources to ensure the model's accuracy (Figure 3). The calibration of the model was executed using data acquired during the rabi 2020–21 wheat growing season, and a spatial analysis mode was employed to determine genetic coefficients. The calibrated model underwent validation by comparing its simulated yield with observed field data, ensuring its reliability for future predictions.

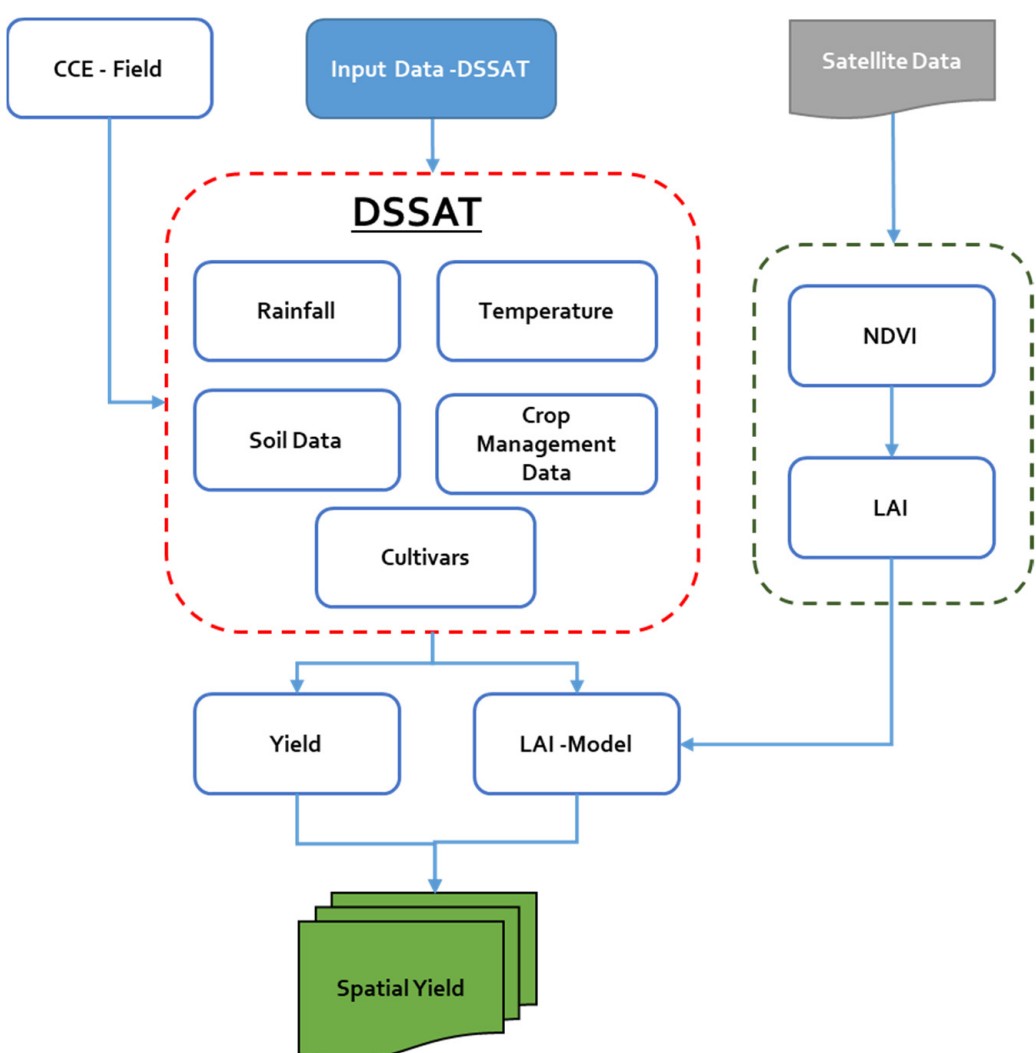

**Figure 3.** Methodology for yield estimation using the DSSAT and its integration with remote sensing.

To enhance the integration of model outcomes with real-world conditions, leaf area index (LAI) values were extracted from NDVI images.

(c)    Semi-Physical Approach

In this study, a comprehensive semi-physical method was employed to assess wheat growth and estimate yield, leveraging multi-temporal datasets from diverse sources (Figure 4) [56]. Sentinel-2 crop classification data, INSAT 3D-supplied daily insolation data at a 1 km spatial resolution, MODIS-contributed information on various parameters and NASA POWER-furnished meteorological data, including temperature extremes, were used. The growing season, spanning December 2020 to May 2021, was the focus of the analysis. Daily insolation data from INSAT 3D, accounting for half of the total insolation, were crucial. Additionally, 8-day composites of the fAPAR from MODIS at a 500 m spatial resolution were utilized. The Net Primary Product (NPP) was then computed at an 8-day

interval using factors like PAR, fAPAR, stress and maximum radiation use efficiency at a 500 m spatial resolution.

$$\text{NPP (g m}^2\text{day1)} = \text{PAR} \times \text{fAPAR} \times \text{RUE} \times \text{stress}$$

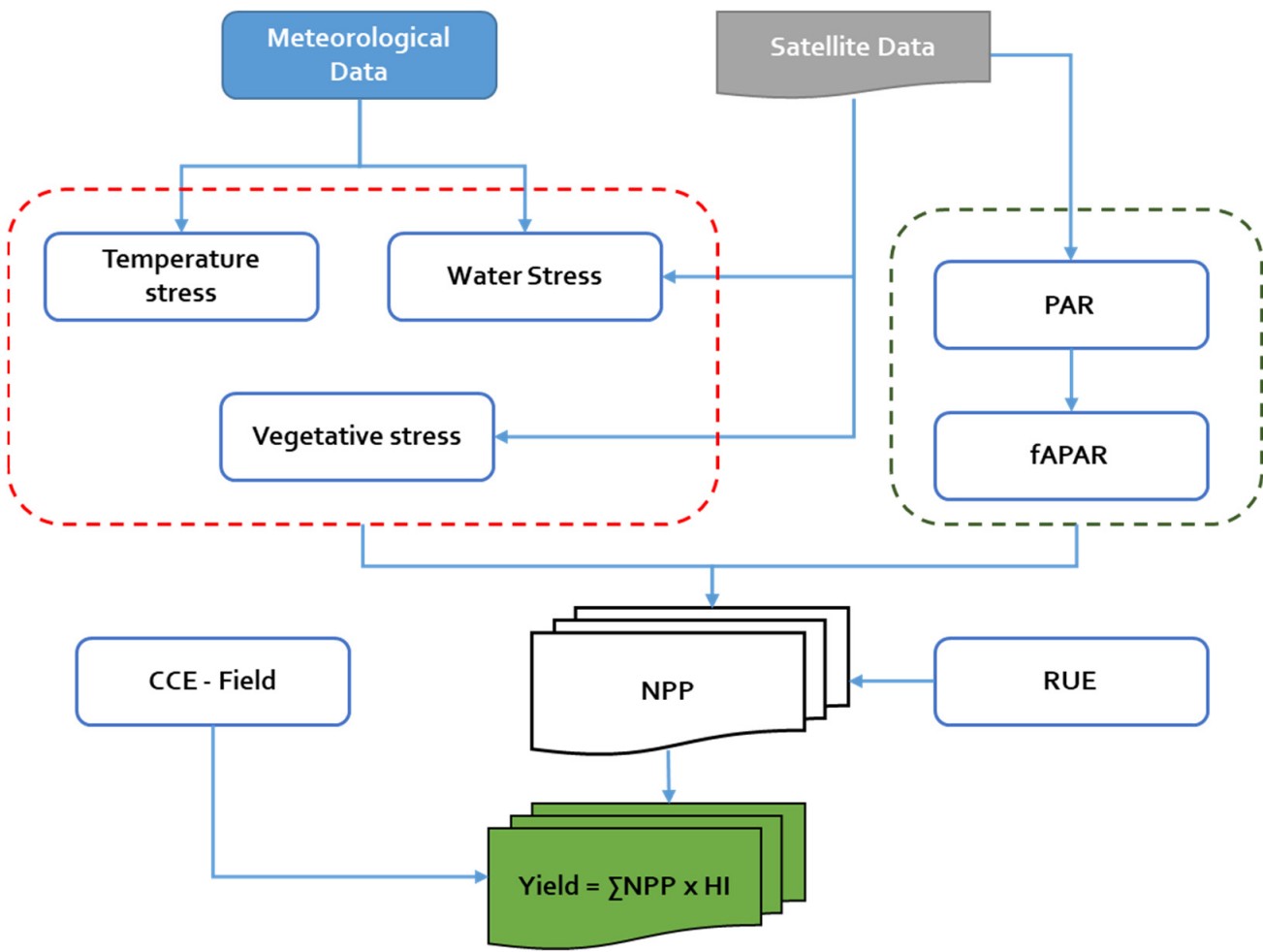

**Figure 4.** Methodology for estimating yield through an SPM.

The radiation use efficiency (RUE) of wheat varies from 1.2 to 2.93 g MJ$^{-1}$, so we have taken the common mean RUE value as 2.06 g MJ$^{-1}$.

The total NPP for the entire wheat growing season was derived from these composite datasets. Wheat yield was subsequently calculated by combining the total NPP with the harvest index obtained from CCEs.

## 3. Results and Discussion

### 3.1. Crop Classification

The classification map of Bareilly reveals a diverse landscape with distinct LULC classes (Figure 5). Among these, wheat cultivation dominates a significant portion of croplands, covering an area of 183,930 ha, whereas other crops contribute nearly 85,939 ha (Table 1).

**Table 1.** The labelled classes and their respective areas across the study area.

| S.no | Class | Area (ha) |
|---|---|---|
| 1 | Wheat | 183,930 |
| 2 | Other Crops | 85,939 |
| 3 | Water bodies | 3195 |
| 4 | Built-up | 13,795 |
| 5 | Other LULCs | 91,523 |

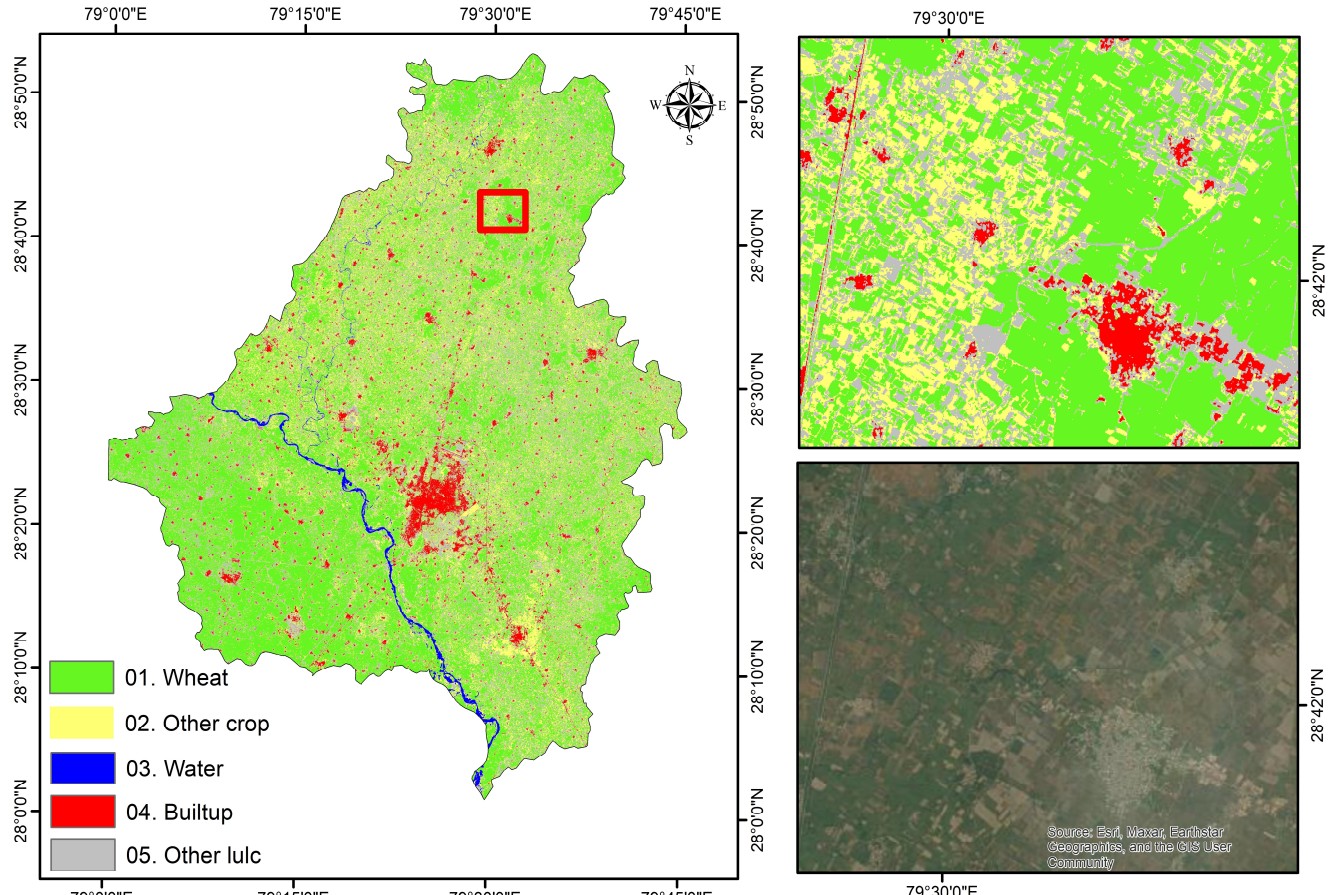

**Figure 5.** Spatial distribution of wheat and other LULCs.

Water bodies, crucial for ecological balance, occupy 3195 ha in the study region. The built-up area, comprising infrastructure and urban spaces, was noted at 13,795 ha. Additionally, the category of "Other LULCs" contains varied land uses, with nearly 91,523 ha. This comprehensive classification provides valuable insights into the spatial distribution of different land cover types, facilitating a complete understanding of Bareilly's diverse and dynamic environment.

A total of 200 ground truth points were randomly collected throughout the crop-growing season in the study area to facilitate validation processes. The accuracy of the wheat area map was evaluated using a confusion matrix, employing ground truth points to distinguish between wheat and non-wheat pixels. The classification accuracy for wheat points was estimated at 95.3%, whereas non-wheat points were classified with an accuracy of 92.0%.

### 3.2. Yield Estimation Using ML Algorithms

Using ML algorithms, the CCE points were utilized as training data against NDVI images to differentiate and predict crop yields across the entire district (Figure 6). The analysis uncovered a notably positive correlation between the NDVI and crop yield, indicating the potential of the NDVI as a reliable indicator for agricultural productivity. Subsequently, pixel-level yields were aggregated to GP levels, offering a more comprehensive understanding of crop performance across different regions/ecosystems.

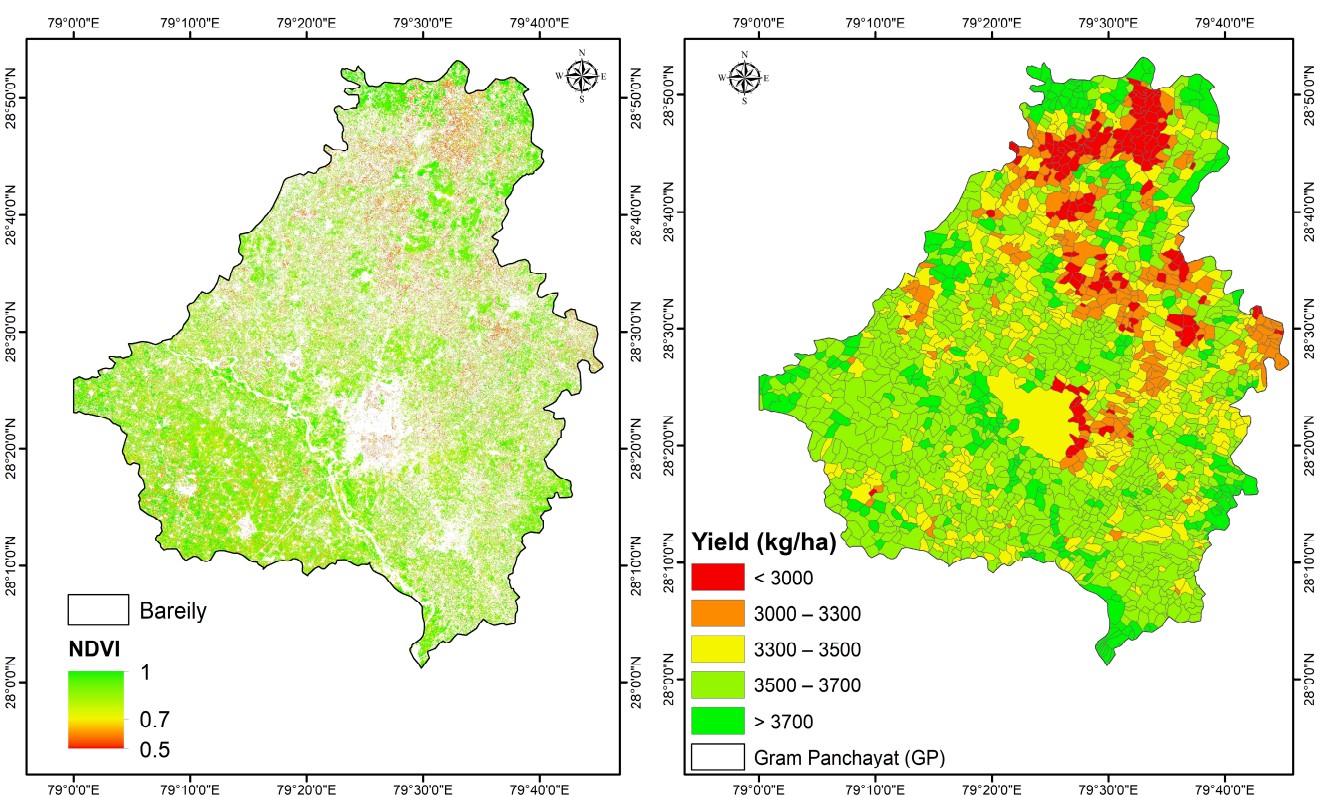

**Figure 6.** Spatial distribution of yield at GP level using ML algorithm.

In the southern part of Bareilly district, a consistent crop yield was observed, suggesting homogeneity in crop health in that area. The predicted yield of the majority of the Gram Panchayats in this region varies from 3500 to 3700 kg/ha. Conversely, the northern part of Bareilly exhibited signs of stress in crop yields, indicating potential challenges or variations in agricultural conditions. The majority of GPs located in this region showed a yield of less than 3300 kg/ha. The GP-level yields displayed noteworthy diversity, fluctuating from 2358 kg/ha to 4180 kg/ha, highlighting the high spatial variability in wheat crop productivity within the district.

### 3.3. Yield Estimation Using DSSAT Crop Simulation Model

The crop simulation model employed in this study serves as a simplified representation of crop growth, considering various influencing factors such as variety, soil, weather and management. Specifically, the CERES-Wheat model was calibrated, tested and validated to simulate wheat yields, taking into account the spatial influence of these input factors.

To generate a spatial leaf area index (LAI) for the corresponding wheat pixels, the classified map was utilized. Initially, NDVI values were extracted from wheat pixels, and considering the noise in the wheat pixels, NDVI thresholds greater than 0.5 were considered. The determination of the remote sensing (RS) LAI involved using the linear equation derived from the correlation between the model's LAI and NDVI. Subsequently,

a spatial wheat yield map was created using the spatial LAI map, establishing a linear relationship between the model's LAI and the model's yield.

The GP-level yield in Bareilly district exhibited a range from 2111 kg/ha to 4628 kg/ha, a variation that was determined through the implementation of the DSSAT model (Figure 7). The model was executed by stratifying areas based on similarities in soil type, weather conditions and cultivation practices. This strategic approach allows for a more precise and localized assessment of agricultural productivity.

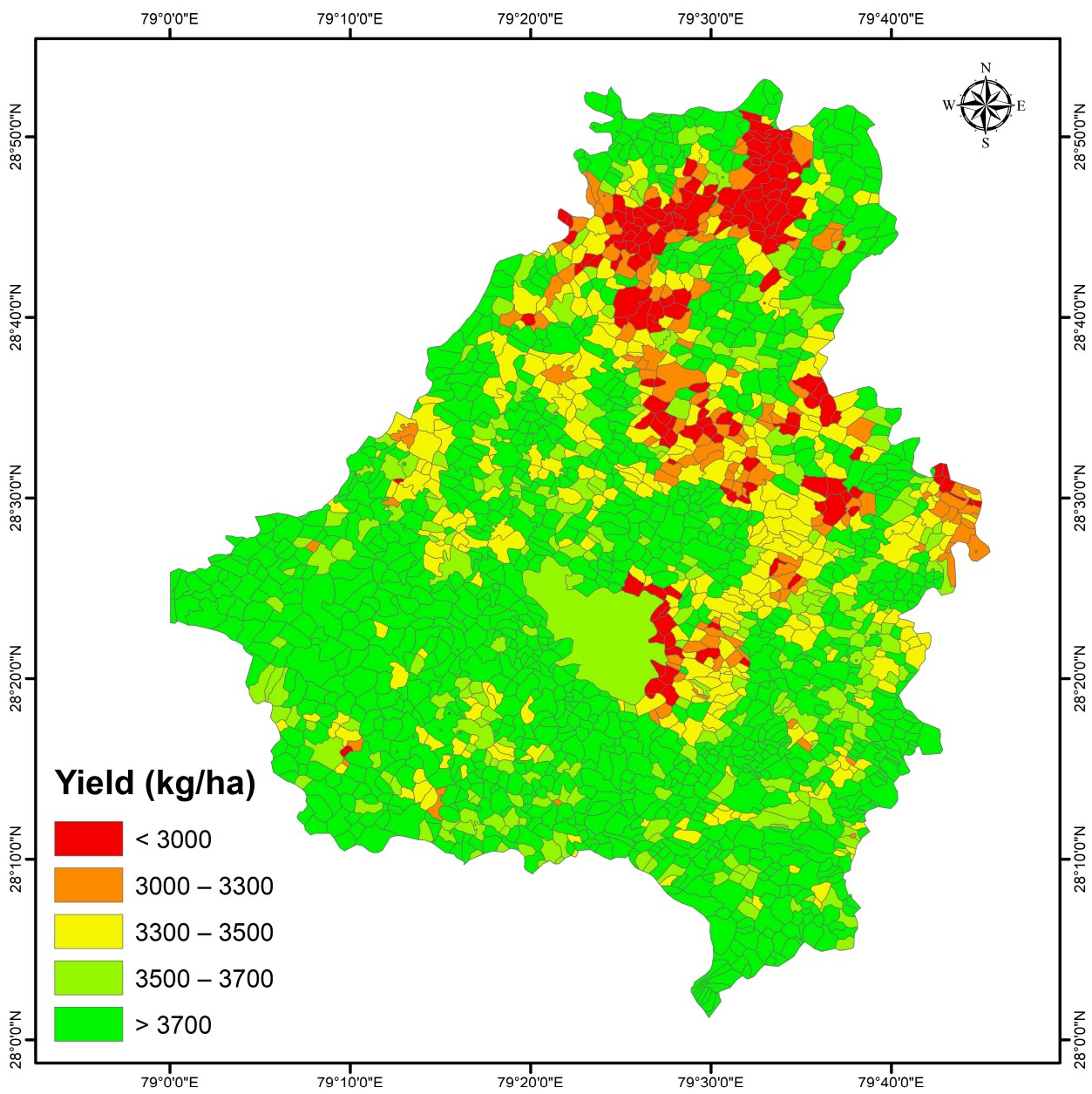

**Figure 7.** Spatial distribution of yield estimation using DSSAT crop simulation model.

These yield calculations are particularly sensitive to the spatial leaf area index (LAI) values, indicating the importance of vegetation density in influencing crop yields. The DSSAT model, by factoring in these spatial variations in the LAI, provides a refined understanding of the relationships between soil, weather and cultivation practices in different regions. The observed range in the GP-level yields underscores the significance of modifying agricultural strategies to specific local conditions, enabling more effective management and optimization of crop production in diverse areas of the district.

### 3.4. Yield Estimation Using Semi-Physical Approach

Employing a semi-physical approach, the GP-level yields in Bareilly district exhibited variability ranging from 2546 kg/ha to 3845 kg/ha (Figure 8). This methodology involves calculating yields based on the conversion of PAR into biomass, taking into account the RUE of the wheat crop, along with the fAPAR.

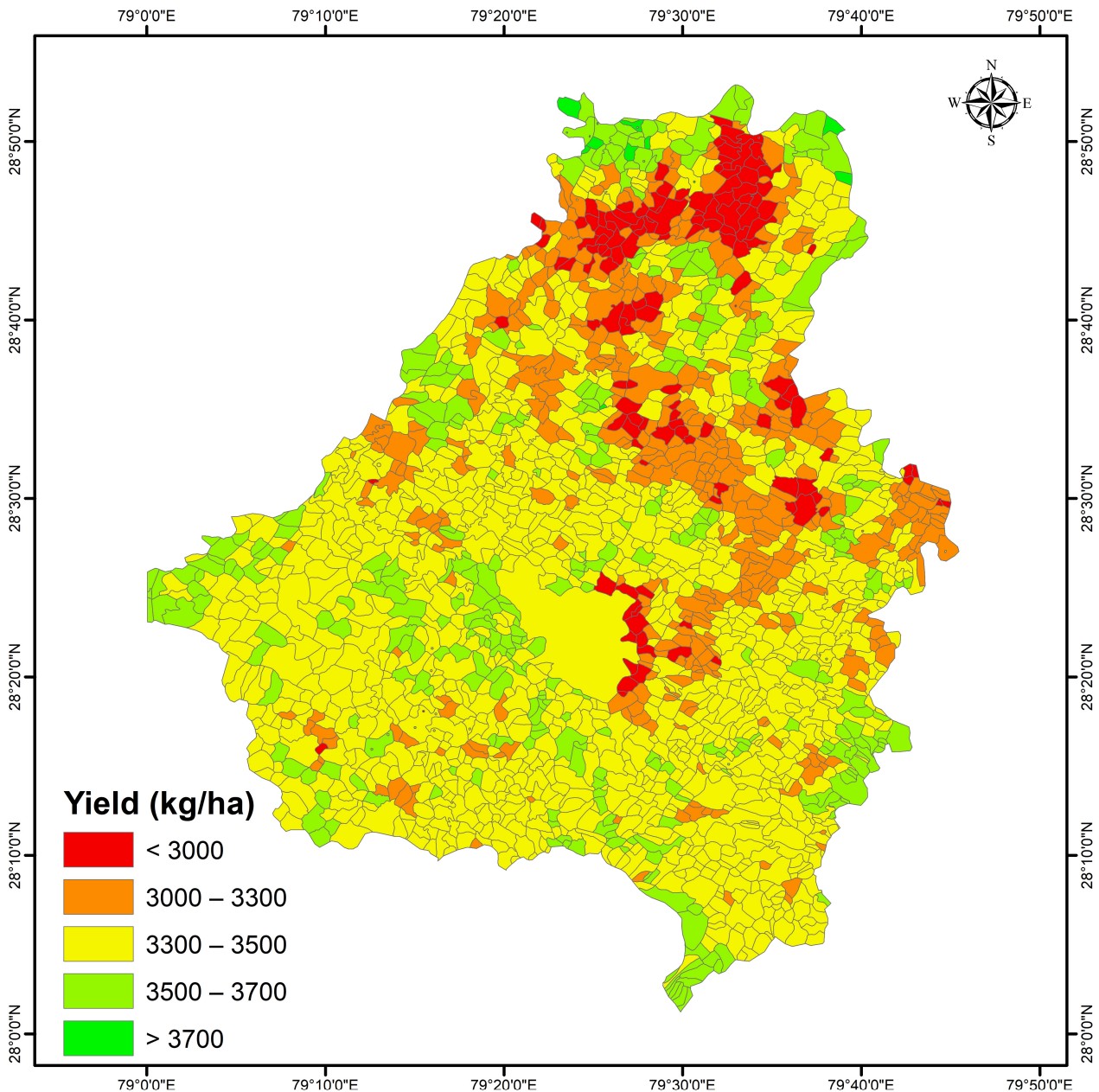

**Figure 8.** Spatial distribution of yield using semi-physical approach.

In the present context, the wheat yield calculation is primarily influenced by PAR and the crop's RUE. PAR serves as a crucial factor in driving photosynthesis, and the RUE of the crop determines how efficiently this radiation is converted into biomass. As a result, variations in these parameters contribute to the observed variation in Gram Panchayat-level yields. This semi-physical approach provides a mechanistic understanding of the factors influencing crop productivity, offering valuable insights into the intricate relationships between radiation, crop physiology and ultimately, yield outcomes.

### 3.5. Comparison between Different Models

The average crop yield estimates for five taluks of Bareilly district, namely Aonla, Baheri, Bareilly, Faridpur and Nawabganj, across three distinct methodologies, namely the SPM, the NDVI and the DSSAT, are presented in the chart below (Figure 9).

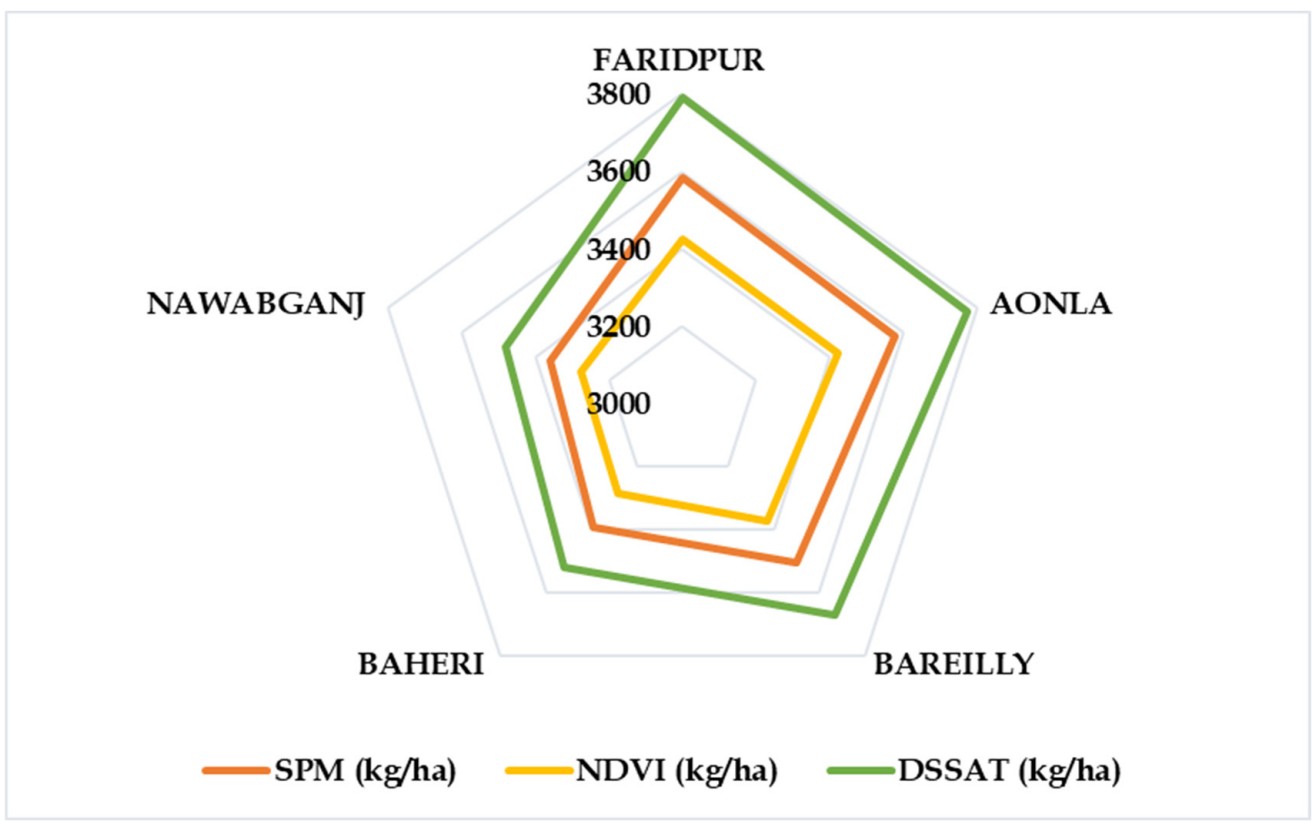

**Figure 9.** Chart showing the yield variations using different models.

The highest average yield is noticed in Faridpur, followed by Aonla, Bareilly and Baheri, and the lowest in Nawabganj. The gap between the yields calculated through these three models is well observed in the above chart.

The performance evaluation of the three distinct crop yield estimation models involves the Root Mean Square Error (RMSE), which provides a measure of prediction accuracy. The striking similarity in their R-squared ($R^2$) values, around 0.82, underscored their comparable explanatory capabilities. For the SPM, an RMSE of 1776 kg/ha was accompanied by an $R^2$ value of nearly 0.82, highlighting its moderate prediction errors and robust correlation with the observed values. Similarly, the NDVI exhibited a slightly higher RMSE at 1907 kg/ha yet maintained a consistent $R^2$ value of approximately 0.82, indicating its effective capacity to explain variations in crop yields. Notably, the DSSAT outperformed the other model with the lowest RMSE of 1605 kg/ha, coupled with a steady $R^2$ value of around 0.82, affirming its superior accuracy and explanatory power (Figure 10). The convergence of the $R^2$ values across all the models suggests a collective strength in capturing the underlying patterns in wheat yield data. While differences in the RMSE highlight varying levels of precision, the model consistently demonstrated a strong correlation between the predicted and observed values.

The study differentiated that the DSSAT model consistently produced the highest average crop yields among the examined methodologies. This performance can be attributed to the model's detailing of input data at the point level. In contrast, both the SPM and the NDVI demonstrated lower average yields. This difference can be attributed to their

dependence on remote sensing data, which, while offering broad coverage, may introduce noise into the analysis, potentially compromising precision.

SPMs exhibit advantages in yield estimation. By incorporating physical principles, these models establish a robust foundation for accurate predictions. Additionally, SPMs prove advantageous in regions with limited data availability, making them applicable in diverse agricultural settings. However, their limitations become apparent through their heavy dependence on accurate input parameters, making predictions susceptible to discrepancies in the data. Furthermore, SPMs may struggle to adapt to dynamic and rapidly changing environmental conditions, potentially impacting their overall adaptability. Similarly, the NDVI leverages satellite-derived vegetation indices, offering a non-invasive and efficient means of estimating crop yields. The NDVI's suitability for large-scale monitoring allows for insights into overall vegetation health. Nonetheless, limitations arise, such as its constraints in assessing vegetation health, potentially overlooking critical factors influencing crop yields. Additionally, the NDVI exhibits sensitivity to atmospheric conditions and susceptibility to interference from non-crop vegetation, impacting the accuracy of its predictions.

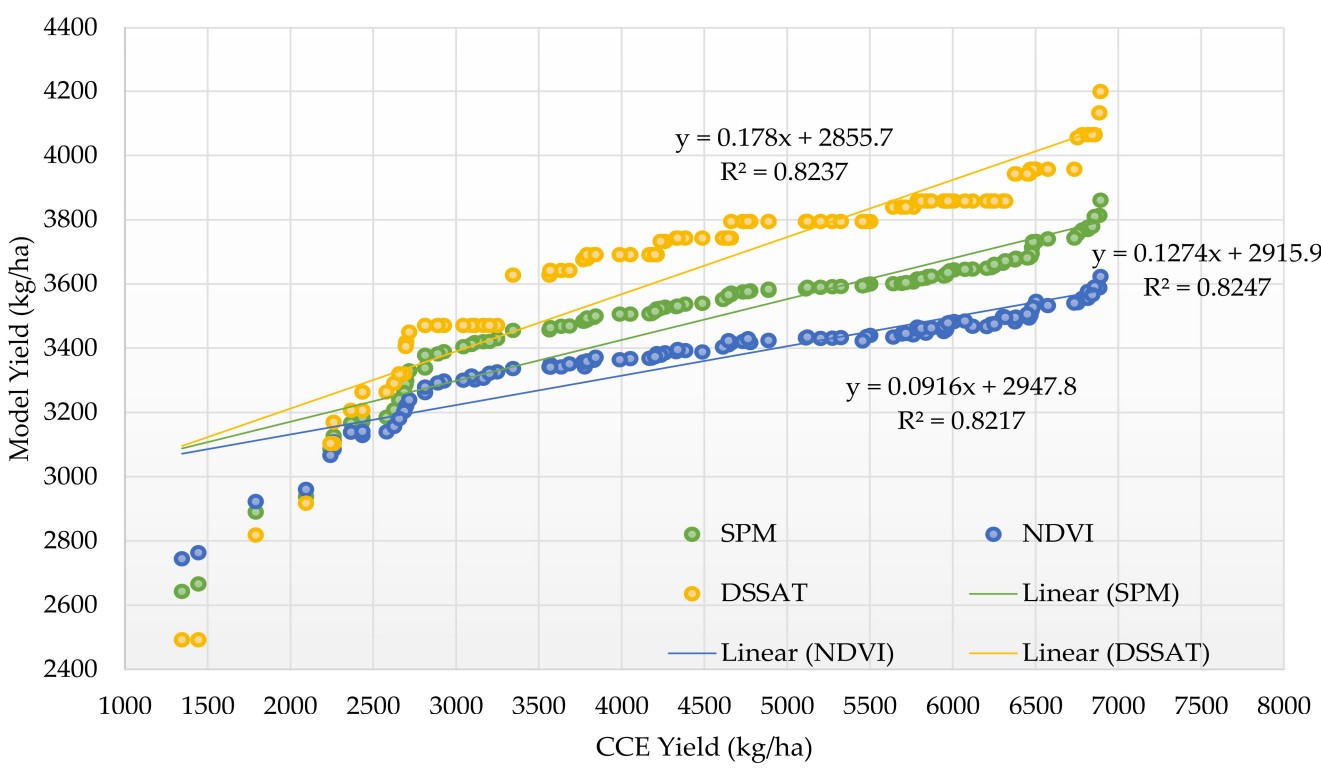

**Figure 10.** Correlation between the CCE data and the model yield.

In the case of the DSSAT, its advantages lie in its comprehensive approach to yield estimation. The DSSAT accounts for complex interactions between soil, weather and crop management, resulting in detailed and comprehensive yield estimates. The model's adaptability is showcased by its suitability for diverse agro-ecological zones, providing flexibility in applications across varying agricultural landscapes. However, the DSSAT comes with a set of limitations, including the requirement for extensive input data, which poses challenges in obtaining accurate and comprehensive information. Moreover, the model is sensitive to inaccuracies in input parameters, potentially affecting the reliability of its yield predictions.

All three models require CCEs to be conducted. However, when compared with the SPM and ML, the DSSAT model demands a more extensive set of parameters, encompassing

soil characteristics, weather variables and crop management practices. This requirement adds a significant level of effort to the implementation of the DSSAT model.

The study underscores the critical role of detailed input data and model calibration in achieving accurate crop yield estimates in near real time. The advantages and limitations associated with each methodology highlight the importance of considering the specific characteristics of the study area when selecting an appropriate modeling approach for crop yield estimation.

## 4. Conclusions

The assessment of wheat crop yields in Bareilly district through the application of diverse methodologies has provided a comprehensive understanding of the agricultural landscape in the target ecosystem. ML algorithms using CCE points and the NDVI revealed a positive correlation between the NDVI and crop yield, explaining the spatial variations across the district.

The research findings clearly articulate how the application of high-end science tools such as remote sensing and data-driven techniques can help in advancing the estimation of wheat crop yields. The calibrated and validated DSSAT model played a crucial role in simulating wheat yields, providing valuable insights into local variations based on factors like soil, weather, cultivation practices, etc. In addition, a semi-physical method, considering factors like PAR, RUE and fAPAR, delved into the detailed aspects influencing crop yields. This approach gave a unique perspective on how environmental factors and crop physiology interact, leading to observed variations in yields at the Gram Panchayat level. Machine learning models demonstrated their effectiveness in homogenous areas with similar cultivars. However, the accuracy of a semi-physical model relies basically on the resolution of the utilized data. The application of the DSSAT model is proficient in predicting yields at specific locations. However, challenges arise when attempting to extrapolate these predictions to encompass a broader study area. The present study contributes valuable insights for policymakers by offering near-real-time, high-resolution crop yield estimates at the local level, thereby facilitating informed decision making to enhance food security at the national level.

By closely comparing machine learning, simulation modeling and semi-physical approaches, this research study offers a comprehensive understanding of the agricultural diversity in wheat cultivation in Bareilly district. The research findings provide a strong foundation for well-informed or evidence-based decision making in wheat crop management, allocating resources, strategizing international trade and finally protecting the interests of small and marginal wheat farmers in the state. The integration of these methodologies signifies the importance of employing diverse tools to capture the complexity of wheat agricultural systems and paves the way for the development of sustainable agricultural strategies in the state.

**Author Contributions:** Conceptualization, M.K.G., R.M.N. and K.C.D.; methodology, M.K.G., R.M.N. and P.P.; software, P.P., P.K.B. and S.G.; validation, M.K.G., P.P., P.K.B. and S.G.; writing—original draft preparation, M.K.G. and P.P.; writing—review and editing, M.K.G., P.P., S.K.D. and V.K.S.; visualization, I.M.; supervision, M.K.G. and K.C.D.; funding acquisition, M.K.G. All authors have read and agreed to the published version of the manuscript.

**Funding:** This research was supported by the MNCFC and ICRISAT. We would like to thank Nagaraju Maila for his support in the ground data collection and logistics arrangements. We are grateful to MNCFC for providing sub-national statistics.

**Data Availability Statement:** The data and materials in this manuscript can be accessed.

**Conflicts of Interest:** No conflicts of interest exist regarding the submission of this manuscript, and the manuscript has been approved by all authors for publication.

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
