# Peer review of "Optimizing Crop Yield Estimation through Geospatial Technology: A Comparative Analysis of a Semi-Physical Model, Crop Simulation, and Machine Learning Algorithms"

_agriengineering, doi:10.3390/agriengineering6010045_

Round 1
Reviewer 1 Report
Comments and Suggestions for Authors
This manuscript compares the results obtained by three different models for predicting crop yield and in addition correlates model predictions to yields obtained in a crop-cutting experiments. The results obtained are important. However, before this study can be accepted for publication, its organization should be ameliorated and discussion of the results should be enhanced.
MAIN CONCERNS
Most of the references are cited in the Introduction section. Material and Methods or Results sections need adequate references. Discussion is lacking. May be Results and Discussion?
The organization of the manuscript should be more consistent. For example, in the Material and Methods and in the Results and Discussion sections the three models rank in the following order: first NDVI, then DSSAT and finally SPM. This is not always the case in other sections of the manuscript. See, for example, lines 15 to 16 and Lines 24-28. Please, be more consistent in organizing the manuscript.
ADDITIONAL COMMENTS
ABSTRACT.
In general, this section should be better organized as before outlined.
What is the meaning of DSSAT? It is the first time DSSAT is used in the manuscript, so previous definition is requested.
What is the objective of this study?
INTRODUCTION
In general, this section should be better organized.
Lines 34 to 37. This statement needs references.
Lines 40 to 43. This statement needs references.
Lines 114-115. In my opinion the objective of this study should be clearly stated. In my opinion the main point is that authors are using different models for predicting crop production. This idea should be more precisely stated.
MATERIAL AND METHODS
References are needed for all the methods described in this section
Lines 119 to 125. Please, describe climate parameters, i.e. precipitation and temperature together, without a mixture with soil and agricultural data. Again, references are needed, in this case for the study area.
Lines 135 and 137. I´m not sure if “geographical location” in these two lines is a redundancy. Geographical location is recorded by two different methods? And again, please, provide references.
Line 137. figure 1 or Figure 1? Please, correct.
Lines 142 to 143.
Please, revise this subsection title: “2.2.1 Mapping wheat growing areas across study area by integrating Sentinel-2 imagery 142 and Ground data”.
For consistency the following is proposed: “2.2.1 Mapping Wheat Growing Areas Across Study Area by Integrating Sentinel-2 Imagery 142 and Ground Data”
RESULTS
I suggest this section should be titled “RESULTS AND DISCUSSION”, as a Discussion section is lacking.
Line 185. Please, revise the title of this subsection.
Line 251. Please, revise “encompasses”, “encompassing”.
Line 262. Please, revise the title of this subsection.
Line 280. Please, revise the title of this subsection.
Figure 10. a) Why are CCE data (kg/ha) much higher than the data estimated by the three models (kg/ha).
b) I’m not sure if the lineal relationship between CCE and model yield is most adequate in this context. What about a semi logarithmic plot or a polynomial relationship? Figure 10 shows that the slope of the relationship is not constant, ut clearly changes with increased yield values.
c) Differences between maximum and minimum yields estimated by the three models used is also an interesting result.
CONLUSIONS
Several sentences in the Conclusion section should be improved for increasing readability.
In addition, Conclusions should contain quantitative data, as quoted in thehat Results section. Also, it is important that Objectives and Conclusions are linked.
Comments on the Quality of English Language
The English Language is grammatically correct. However, English editing is recommended mainly for increasing readability and also for minor grammar checking.
Thus, several sentences may need tightening up to enhance its readability and comprehensibility. Next I’m providing examples of unclear sentences or sentences with ambiguous terms or ambiguous phrasing. Please, authors should be aware that this Language review is not exhaustive, and therefore a thoroughly review of all the sections is recommended.
Line 25. What is the meaning of contingent, in this context?
Line 28. I wonder if “provide” may be more adequate than “contribute
Line 28. “This research contributes valuable insights for policymakers”, is unclear and difficult to understand.
Etc.
Etc.
And again, several sentences in the Conclusion section should be improved for increasing readability.
Author Response
Reviewer-1
- This manuscript compares the results obtained by three different models for predicting crop yield and in addition correlates model predictions to yields obtained in a crop-cutting experiments. The results obtained are important. However, before this study can be accepted for publication, its organization should be ameliorated and discussion of the results should be enhanced.
Our response: Thanks for reviewing our paper. We have revised our paper based on your valuable suggestions and comments. We express our gratitude for the insightful comments you provided. Your constructive feedback and thoughtful suggestions are highly appreciated and will play a crucial role in refining and improving the overall quality of our manuscript. we sincerely thank you for your valuable input.
MAIN CONCERNS
- Most of the references are cited in the Introduction section. Material and Methods or Results sections need adequate references. Discussion is lacking. May be Results and Discussion? The organization of the manuscript should be more consistent. For example, in the Material and Methods and in the Results and Discussion sections the three models rank in the following order: first NDVI, then DSSAT and finally SPM. This is not always the case in other sections of the manuscript. See, for example, lines 15 to 16 and Lines 24-28. Please, be more consistent in organizing the manuscript.
Our response: Thanks for your suggestions and comments. We have revised the manuscript and followed consistency (ML, DSSAT and SPM) throughout (In the abstract and Introduction section). We have discussed most of the methods available in the introduction section and cited the references whereas in the methods section, we have used citations for crop classification and yield estimation. In discussion section, we have discussed every yield estimation model and also compared three models in one section.
ADDITIONAL COMMENTS: ABSTRACT.
- In general, this section should be better organized as before outlined.
Our response: Thanks for your suggestion, we have revised few sentences for better understanding.
“Machine learning models exhibit effectiveness in homogenous areas with similar cultivars, while the accuracy of semi-physical models depends upon the resolution of the utilized data. The DSSAT model is effective in predicting yields at specific locations, but faces difficulties when trying to extend these predictions to cover a larger study area.”
- What is the meaning of DSSAT? It is the first time DSSAT is used in the manuscript, so previous definition is requested.
Our response: Thanks for your comment. We have used the definition of DSSAT at first.
“Decision Support System for Agrotechnology Transfer (DSSAT) crop model”
- What is the objective of this study?
Our response: Thanks for comment. We have revised the objectives for better understanding. “This study aimed to compare the effectiveness of using remote sensing in conjunction with DSSAT, ML, DSSAT and SPM. The major objectives include mapping wheat grown areas in the study area followed by yield estimation using ML, DSSAT and SPM, and comparing the models.”
INTRODUCTION
- In general, this section should be better organized.
Our response: Thanks for your comment, we have revised the few paragraphs for better understanding. From 31 to 131.
- Lines 34 to 37. This statement needs references.
Our response: Added reference for the statement in line 39 “Benami, E., Jin, Z., Carter, M.R., Ghosh, A., Hijmans, R.J., Hobbs, A., Kenduiywo, B. and Lobell, D.B., 2021. Uniting remote sensing, crop modelling and economics for agricultural risk management. Nature Reviews Earth & Environment, 2(2), pp.140-159.”
- Lines 40 to 43. This statement needs references.
Our response: Added reference for the statement in line 44 “Ramadas, S., Kumar, T.K. and Singh, G.P., 2019. Wheat production in India: Trends and prospects. In Recent advances in grain crops research. IntechOpen.”
- Lines 114-115. In my opinion the objective of this study should be clearly stated. In my opinion the main point is that authors are using different models for predicting crop production. This idea should be more precisely stated.
Our response: Thanks for the comment. We have revised the objectives for better understanding. Lines 128-131 “This study aimed to compare the effectiveness of using remote sensing in conjunction with DSSAT, ML, DSSAT and SPM. The major objectives include mapping wheat grown areas in the study area followed by yield estimation using ML, DSSAT and SPM, and comparing the models.”
MATERIAL AND METHODS
- References are needed for all the methods described in this section
Our response: Thanks for your comment. We have added reference for every method in subsection 2.2.1 and 2.3
- Lines 119 to 125. Please, describe climate parameters, i.e. precipitation and temperature together, without a mixture with soil and agricultural data. Again, references are needed, in this case for the study area.
Our response: Thanks for your comment. The temperature and precipitation of study has given in study area section. We have added reference for the climate profile in line 137.
- Lines 135 and 137. I´m not sure if “geographical location” in these two lines is a redundancy. Geographical location is recorded by two different methods? And again, please, provide references.
Our response: Thanks for the comment. Here geographical location indicates the latitude and longitude for specific location in WGS84 projection.
- Line 137. figure 1 or Figure 1? Please, correct.
Our response: Thanks for the comment. Revised “Figure 1”.
- Lines 142 to 143. Please, revise this subsection title: “2.2.1 Mapping wheat growing areas across study area by integrating Sentinel-2 imagery 142 and Ground data”. For consistency the following is proposed: “2.2.1 Mapping Wheat Growing Areas Across Study Area by Integrating Sentinel-2 Imagery and Ground Data”
Our response: Thanks for the comment. We have revised the subsection titles.
RESULTS
- I suggest this section should be titled “RESULTS AND DISCUSSION”, as a Discussion section is lacking.
Our response: Thanks for your comment. We have revised the section name to “Results and Discussion”
- Line 185. Please, revise the title of this subsection.
Our response: we have revised the title for consistency
- Line 251. Please, revise “encompasses”, “encompassing”.
Our response: we have revised the sentence for better understanding
- Line 262. Please, revise the title of this subsection.
Our response: we have revised the title for consistency
- Line 280. Please, revise the title of this subsection.
Our response: we have revised the title for consistency
- Figure 10. a) Why are CCE data (kg/ha) much higher than the data estimated by the three models (kg/ha). b) I’m not sure if the lineal relationship between CCE and model yield is most adequate in this context. What about a semi logarithmic plot or a polynomial relationship? Figure 10 shows that the slope of the relationship is not constant, ut clearly changes with increased yield values. c) Differences between maximum and minimum yields estimated by the three models used is also an interesting result.
Our response: Thanks for your suggestions and comments. Here are our responses
- a) Crop Cutting Experiments (CCE) involve field measurements taken from a 5m x 5m sample, extrapolated to hectares, and aggregated to Gram Panchayat (GP) level. This method may not capture the entire stress in the field. In contrast, remote sensing considers stress across the GP, providing a more comprehensive view. Consequently, CCE data may show higher values in some GPs compared to the model.
- b) Our study utilized a basic approach for yield estimation, and we did not extend it to a polynomial level.
- c) The comparison of average yields from the three models was conducted at the GP and taluk levels. This indirect approach inherently considers the minimum and maximum yields within the area.
- Several sentences in the Conclusion section should be improved for increasing readability. In addition, Conclusions should contain quantitative data, as quoted in the that Results section. Also, it is important that Objectives and Conclusions are linked.
Our response: Thanks for your comment. We have revised the objectives as suggested and quantitative data was discussed in section 3.5. Based on the suggestions given, the objectives and conclusions are linked.
- Comments on the Quality of English Language The English Language is grammatically correct. However, English editing is recommended mainly for increasing readability and also for minor grammar checking. Thus, several sentences may need tightening up to enhance its readability and comprehensibility. Next I’m providing examples of unclear sentences or sentences with ambiguous terms or ambiguous phrasing. Please, authors should be aware that this Language review is not exhaustive, and therefore a thoroughly review of all the sections is recommended.
Our response: Thanks for your comment. This manuscript has gone through English review with an independent reader.
- Line 25. What is the meaning of contingent, in this context?
Our response: Thanks for the comment. We have revised the sentence for better understanding.
- Line 28. I wonder if “provide” may be more adequate than “contribute
Our response: Thanks for the comment. We have revised the sentence as suggested for better understanding.
- Line 28. “This research contributes valuable insights for policymakers”, is unclear and difficult to understand.
Our response: Thanks for your comment. “This study helps policymakers know about different crops grown and their yield variations in that area and they can take necessary decisions”
And again, several sentences in the Conclusion section should be improved for increasing readability.
Our response: Thanks for your comment. We have revised sentences in conclusion section for better readability.
Reviewer 2 Report
Comments and Suggestions for Authors
Starting from the problem of crop yield prediction, this paper selects the Bareilly district in Uttar Pradesh, India, as the research object. Machine Learning Algorithms and DSSAT crop simulation models were compared and analyzed from gram panchayat crop yields model) and Semi-Physical Model (semi-physical Model) prediction accuracy, advantages and disadvantages of the three methods, the reasons for some differences in the predicted results of the three methods, and which method should be adopted for different regions. I think the results presented in this article are very valuable and the conclusions drawn are very attractive.
However, there are some flaws in the article. In my opinion, there are mainly these:
1. In the research scope of the second part, the period of the research object can be explained.
2. The introduction of machine learning algorithms in the second part should explain why NDVI can be used as a proxy variable for crop yield.
3. The description of remote sensing eigenvalues can be appropriately added to the machine learning algorithm and semi-physical model in the second part.
4. It can be seen from the third part that the prediction results of DSSAT crop model are the best, so specific control tests can be conducted on the five input data of DSSAT in the second part, so as to further draw results and put forward policy suggestions.
5. The format of the text section under the 3.5 subheading is different from the previous one.
6, the article directly from 3.Results to 5.Conclusions should be changed from 5 to 4.
Author Response
Reviewer -2
Starting from the problem of crop yield prediction, this paper selects the Bareilly district in Uttar Pradesh, India, as the research object. Machine Learning Algorithms and DSSAT crop simulation models were compared and analyzed from gram panchayat crop yields model) and Semi-Physical Model (semi-physical Model) prediction accuracy, advantages and disadvantages of the three methods, the reasons for some differences in the predicted results of the three methods, and which method should be adopted for different regions. I think the results presented in this article are very valuable and the conclusions drawn are very attractive.
Our response: We express our gratitude for the insightful comments you provided. Your constructive feedback and thoughtful suggestions are highly appreciated and will play a crucial role in refining and improving the overall quality of our manuscript. we sincerely thank you for your valuable input.
However, there are some flaws in the article. In my opinion, there are mainly these:
- In the research scope of the second part, the period of the research object can be explained.
Our response: Thanks for the comment. The period of the research object is rabi 2020-21 i.e. November 2020 to March 2021. In lines 147 and 183-184
- The introduction of machine learning algorithms in the second part should explain why NDVI can be used as a proxy variable for crop yield.
Our response: Thanks for the comments. Inserted in section “NDVI is a reliable proxy for crop yield as it reflects the health and growth of vegetation. Derived from remote sensing data, it correlates with factors like chlorophyll content and leaf area index, providing insights into crop development. Its non-destructive and cost-effective nature, along with high spatial and temporal resolution, makes NDVI a valuable tool for monitoring large agricultural areas. It helps identify stressors and diseases, enabling timely interventions. However, for more accurate predictions, it's essential to integrate NDVI data with other contextual information such as weather and soil conditions.”
- The description of remote sensing eigenvalues can be appropriately added to the machine learning algorithm and semi-physical model in the second part.
Our response: Thanks for the comment. We have used Random forest machine algorithm for estimating yield with the help of CCE and remote sensing NDVI and in semi physical model, remote sensing PAR is converted to yield with the help of fAPAR and Harvest Index.
- It can be seen from the third part that the prediction results of DSSAT crop model are the best, so specific control tests can be conducted on the five input data of DSSAT in the second part, so as to further draw results and put forward policy suggestions.
Our response: Thanks for your suggestion, we have included the suggestions “Machine learning models demonstrate effectiveness in homogenous areas with similar cultivars, whereas the accuracy of semi-physical models relies on the resolution of the utilized data. While the DSSAT model is proficient in predicting yields at specific locations, challenges arise when attempting to extrapolate these predictions to encompass a broader study area. This research contributes valuable insights for policymakers by offering near real-time, high-resolution crop yield estimates at the local level, thereby facilitating informed decision-making to enhance food security.”
- The format of the text section under the 3.5 subheading is different from the previous one.
Our response: Thanks for the comments. We have revised it to consistent format.
- the article directly from 3. Results to 5. Conclusions should be changed from 5 to 4.
Our response: Thanks for noticing. We have revised it.
Reviewer 3 Report
Comments and Suggestions for Authors
The article "Optimizing Crop Yield Estimation through Geospatial Technology: A Comparative Analysis of Semi-Physical Model, Crop Simulation, and Machine Learning Algorithms" presents a research of the effectiveness of using geospatial technologies to assess crop yields.
This study focuses on determining the critical importance of accurate crop yield information for national food security and export considerations. The paper describes the use of technologies such as the DSSAT model, semi-physical models and machine learning algorithms. The study combines Sentinel time series data and ground data to create comprehensive maps of crop types.
The results of the study showed that the assessment of crop yields in Bareilly district using various techniques allowed obtaining a comprehensive understanding of the agricultural landscape. The results highlight the importance of remote sensing and data based methods for determining the health and productivity of crops.
These methods offered a unique perspective on the interaction of environmental factors and crop physiology, which led to yield fluctuations at the GP level. Together, the combination of ML, simulation modeling and semi-physical approaches gave a complete picture of the dynamics of agriculture in Bareilly. This comprehensive understanding, covering spatial, modeled and mechanistic aspects, provides a reliable basis for making informed decisions in the field of crop management, resource allocation and optimization of farming methods to increase the overall crop yield in the district. The integration of these methodologies demonstrates the importance of using a variety of tools to account for the complexity of agricultural systems and pave the way for sustainable and effective agricultural strategies.
The article is well structured, with a clear introduction, methodology, the main part, results. The authors present the results of their experiments in a clear and understandable form, with figures and graphs that effectively illustrate the main conclusions. The introduction provides a broad overview of research papers on the topic of Crop Yield Estimation.
However, despite the advantages, there are comments on the design and content of the study.
1.On page 4, in lines 131 and 137, "figure" is written with a small letter
2. Abbreviations in the text should be made in the same design style. In the text, abbreviations are presented in different design styles (For example, in line 64 "Spectral Matching Techniques" each word is written with a capital letter, and in line 70 "Photosynthetically active radiation" with a small one);
3. The axes are not signed in Figure 1;
4. For ease of perception, it is recommended to make a notation table indicating the variables and their values;
5. In Table 1, the "Class" column has duplicated information about numbering;
6. The article does not provide a justification for the choice of approximation lines in terms of the physical properties of the object in Figure 10. The equations of the first-order lines are shown, but it is not shown that they best approximate the function;
7. The article shows the use of ML algorithms to solve the problem. To understand the article, it is necessary to add a description of the ML architecture, justification for choosing the ML architecture, and description of its properties.
Given the overall positive result in the article "Optimizing Crop Yield Estimation through Geospatial Technology: A Comparative Analysis of Semi-Physical Model, Crop Simulation, and Machine Learning Algorithms", it is recommended to reconsider the article for publication in the journal AgriEngineering after major revision and correction of comments.
Comments on the Quality of English Language
The article is written at a fairly good level of English. The text is clear. There are small comments, but this does not prevent readers from understanding the content.
Author Response
Reviewer- 3
The article "Optimizing Crop Yield Estimation through Geospatial Technology: A Comparative Analysis of Semi-Physical Model, Crop Simulation, and Machine Learning Algorithms" presents a research of the effectiveness of using geospatial technologies to assess crop yields.
This study focuses on determining the critical importance of accurate crop yield information for national food security and export considerations. The paper describes the use of technologies such as the DSSAT model, semi-physical models and machine learning algorithms. The study combines Sentinel time series data and ground data to create comprehensive maps of crop types.
The results of the study showed that the assessment of crop yields in Bareilly district using various techniques allowed obtaining a comprehensive understanding of the agricultural landscape. The results highlight the importance of remote sensing and data based methods for determining the health and productivity of crops.
These methods offered a unique perspective on the interaction of environmental factors and crop physiology, which led to yield fluctuations at the GP level. Together, the combination of ML, simulation modeling and semi-physical approaches gave a complete picture of the dynamics of agriculture in Bareilly. This comprehensive understanding, covering spatial, modeled and mechanistic aspects, provides a reliable basis for making informed decisions in the field of crop management, resource allocation and optimization of farming methods to increase the overall crop yield in the district. The integration of these methodologies demonstrates the importance of using a variety of tools to account for the complexity of agricultural systems and pave the way for sustainable and effective agricultural strategies.
The article is well structured, with a clear introduction, methodology, the main part, results. The authors present the results of their experiments in a clear and understandable form, with figures and graphs that effectively illustrate the main conclusions. The introduction provides a broad overview of research papers on the topic of Crop Yield Estimation.
Our response: We express our gratitude for the insightful comments you provided. Your constructive feedback and thoughtful suggestions are highly appreciated and will play a crucial role in refining and improving the overall quality of our manuscript. we sincerely thank you for your valuable input.
However, despite the advantages, there are comments on the design and content of the study.
- On page 4, in lines 131 and 137, "figure" is written with a small letter
Our response: Thanks for the comment. We have revised it.
- Abbreviations in the text should be made in the same design style. In the text, abbreviations are presented in different design styles (For example, in line 64 "Spectral Matching Techniques" each word is written with a capital letter, and in line 70 "Photosynthetically active radiation" with a small one);
Our response: Thanks for the comment. We have taken care of it.
- The axes are not signed in Figure 1;
Our response: Thanks for noticing. The axes represent the latitude and longitudes of respective maps.
- For ease of perception, it is recommended to make a notation table indicating the variables and their values;
Our response: Thanks for the suggestion. We have given individual sub section for each methods and respective variables for better understanding. If we insert table, the table shows the duplicate.
- In Table 1, the "Class" column has duplicated information about numbering;
Our response: Thanks for the comment. We have revised the table
|
S.no |
Class |
Area (ha) |
|
1 |
Wheat |
183930 |
|
2 |
Other Crop |
85939 |
|
3 |
Water bodies |
3195 |
|
4 |
Built-up |
13795 |
|
5 |
Other LULC |
91523 |
- The article does not provide a justification for the choice of approximation lines in terms of the physical properties of the object in Figure 10. The equations of the first-order lines are shown, but it is not shown that they best approximate the function;
Our response: Thanks for the comment. The study mainly focused on different basic approaches in yield estimation. The three methods can estimate the yield accurately providing accurate and high resolution data. So, we have analyzed it with the first order.
- The article shows the use of ML algorithms to solve the problem. To understand the article, it is necessary to add a description of the ML architecture, justification for choosing the ML architecture, and description of its properties.
Our response: Thanks for the comment. The basic structure of the ML algorithm is discussed in the section. “This approach uses the random forest algorithm to effectively analyze the complex relationships within the stacked NDVI layers, enabling accurate predictions based on the provided CCE training data.”
- Given the overall positive result in the article "Optimizing Crop Yield Estimation through Geospatial Technology: A Comparative Analysis of Semi-Physical Model, Crop Simulation, and Machine Learning Algorithms", it is recommended to reconsider the article for publication in the journal AgriEngineering after major revision and correction of comments.
Our response: Thanks for reviewing our paper. We appreciate your recommendations. Your comments and responses have improved our paper better.
Round 2
Reviewer 1 Report
Comments and Suggestions for Authors
The revised version of this manuscript has been improved. Authors included most of the recomendations I made in m review of the first version.
Next, I'm sending some additionalminor comments.
1) For consistence, the sentence "The district's temperature variations, ranging from hot summers exceeding 40 degrees Celsius to cool winters between 4 and 20 degrees Celsius, contribute to the cultivation of a diverse range of crops" shold be placed at line 126.
This way, precipitation and temperature data are associated., i.e. temperature follows precipitation.
2) Headings 3.4 and 3.5
3.4. Should be "Yield Estimation using Semi Physical Approach·
Also heading 3.5 should be corrected
3) In my opinion discssion of Figure 10 should be enhanced-
Comments on the Quality of English Language
The English language has been improved. However, again, text editing is recommended.
1) For consistence, the sentence "The district's temperature variations, ranging from hot summers exceeding 40 degrees Celsius to cool winters between 4 and 20 degrees Celsius, contribute to the cultivation of a diverse range of crops" shold be placed at line 126.
This way, precipitation and temperature data are associated., i.e. temperature follows precipitation.
2) Headings 3.4 and 3.5
3.4. Should be "Yield Estimation using Semi Physical Approach·
Also heading 3.5 should be corrected
3) In my opinion discssion of Figure 10 should be enhanced-
Author Response
Reviewer-1
The revised version of this manuscript has been improved. Authors included most of the recommendations I made in my review of the first version.
Our response: Thank you for accepting our responses, and we're grateful for the constructive comments that have significantly enhanced our manuscript. Thank you for conducting a second review; we have incorporated our responses in a structured and thoughtful manner, addressing each comment individually.
Next, I'm sending some additional minor comments.
1) For consistence, the sentence "The district's temperature variations, ranging from hot summers exceeding 40 degrees Celsius to cool winters between 4 and 20 degrees Celsius, contribute to the cultivation of a diverse range of crops" shold be placed at line 126.
This way, precipitation and temperature data are associated., i.e. temperature follows precipitation.
Our response: Thanks for your suggestion, we have revised the sentences. Lines 125-129
2) Headings 3.4 and 3.5; 3.4. Should be "Yield Estimation using Semi Physical Approach·; Also heading 3.5 should be corrected
Our response: Thanks for noticing. We have revised the title for consistency.
3) In my opinion discussion of Figure 10 should be enhanced
Our response: Thanks for your comment. We have enhanced the Figure 10 with more detailing.
4) The English language has been improved. However, again, text editing is recommended.
Our response: Thank you for your suggestion. We have implemented text editing throughout the paper wherever necessary.
Reviewer 3 Report
Comments and Suggestions for Authors
The authors have done a good revision of the article. We have corrected all the points that were in the comments. The article has indeed been improved and the necessary information has been added. The authors have done a good job of improving the article and it can be accepted in this form.
Author Response
The authors have done a good revision of the article. We have corrected all the points that were in the comments. The article has indeed been improved and the necessary information has been added. The authors have done a good job of improving the article and it can be accepted in this form.
Our response: We express our sincere gratitude for acknowledging and accepting our responses. Your recommendation for the approval of our paper is invaluable to us. Thank you for considering and endorsing our contribution